# Direct limits for scalar field dark matter from a gravitational-wave detector

Sander M. Vermeulen[1], Philip Relton[1], Hartmut Grote[1✉], Vivien Raymond[1], Christoph Affeldt[2], Fabio Bergamin[2], Aparna Bisht[2], Marc Brinkmann[2], Karsten Danzmann[2], Suresh Doravari[2], Volker Kringel[2], James Lough[2], Harald Lück[2], Moritz Mehmet[2], Nikhil Mukund[2], Séverin Nadji[2], Emil Schreiber[2], Borja Sorazu[3], Kenneth A. Strain[2,3], Henning Vahlbruch[2], Michael Weinert[2], Benno Willke[2] & Holger Wittel[2]

The nature of dark matter remains unknown to date, although several candidate particles are being considered in a dynamically changing research landscape[1]. Scalar field dark matter is a prominent option that is being explored with precision instruments, such as atomic clocks and optical cavities[2-8]. Here we describe a direct search for scalar field dark matter using a gravitational-wave detector, which operates beyond the quantum shot-noise limit. We set new upper limits on the coupling constants of scalar field dark matter as a function of its mass, by excluding the presence of signals that would be produced through the direct coupling of this dark matter to the beam splitter of the GEO600 interferometer. These constraints improve on bounds from previous direct searches by more than six orders of magnitude and are, in some cases, more stringent than limits obtained in tests of the equivalence principle by up to four orders of magnitude. Our work demonstrates that scalar field dark matter can be investigated or constrained with direct searches using gravitational-wave detectors and highlights the potential of quantum-enhanced interferometry for dark matter detection.

Laser interferometers have very high sensitivity to minute length changes and have facilitated many gravitational-wave detections over the past few years[9,10]. In addition to their merit in astrophysics, the detection of gravitational waves has also shed light on fundamental physics questions and several links may exist between gravitational waves and dark matter[11]. Owing to their excellent sensitivity at or beyond quantum limits, gravitational-wave detectors (or precision interferometers of a similar type) can be used directly for fundamental physics, without the mediation of gravitational waves. Examples include a possible search for vacuum birefringence[12] and the search for signatures of quantum gravity[13-15]. Several ideas have been put forward as to how different candidates of dark matter can directly couple to gravitational-wave detectors, ranging from scalar field dark matter[4,16] to dark photon dark matter[17] and to clumpy dark matter coupling gravitationally or through an additional Yukawa force[18]. Upper limits on dark photon dark matter have already been set in a small mass band using data from the first observational run (O1) of the Advanced LIGO gravitational-wave detectors[19].

In this work, we conduct a direct search for scalar field dark matter using a gravitational-wave detector, the quantum-enhanced GEO600 interferometer, and set new upper limits on the parameters of such dark matter.

## Theory

Models of weakly coupled low-mass (≪1 eV) scalar fields predict that such particles could be produced in the early Universe through a vacuum misalignment mechanism and would manifest as a coherently oscillating field[2,4],

$$\phi(t, r) = \phi_0 \cos(\omega_\phi t - \mathbf{k}_\phi \cdot \mathbf{r}), \qquad (1)$$

where $\omega_\phi = (m_\phi c^2)/\hbar$ is the angular Compton frequency and $\mathbf{k}_\phi = (m_\phi \mathbf{v}_{obs})/\hbar$ is the wave vector, with $m_\phi$ the mass of the field and $\mathbf{v}_{obs}$ the velocity relative to the observer. The amplitude of the field can be set as $\phi_0 = (\hbar \sqrt{2\rho_{local}})/(m_\phi c)$, under the assumption that this scalar field constitutes the local dark matter density $\rho_{local}$ (ref. [20]).

Moreover, these models predict that such dark matter would be trapped and virialized in gravitational potentials, leading to a Maxwell–Boltzmann-like distribution of velocities $\mathbf{v}_{obs}$ relative to an observer. As non-zero velocities produce a Doppler shift of the observed dark matter field frequency, this virialization results in the dark matter field having a finite coherence time or, equivalently, a spread in observed frequency (linewidth)[17,21]. The linewidth is determined by the virial velocity, which is given by the depth of the gravitational potential. For dark matter trapped in the galactic gravity potential, as in the standard galactic dark matter halo model, the expected linewidth is $\Delta\omega_{obs}/\omega_{obs} \sim 10^{-6}$. Certain kinds of scalar particles, such as relaxion dark matter[22,23], may also form gravitationally bound objects and be captured in the gravitational potential of the Earth or the Sun, producing a local dark matter overdensity where the field has a much narrower linewidth[24]. The observed dark matter frequency is further modulated by the

[1]Gravity Exploration Institute, Cardiff University, Cardiff, UK. [2]Max Planck Institute for Gravitational Physics and Leibniz University Hannover, Hannover, Germany. [3]School of Physics & Astronomy, University of Glasgow, Glasgow, UK. ✉e-mail: groteh@cardiff.ac.uk

motion of the Earth with respect to the centre of mass of the local dark matter.

Scalar field dark matter could couple to the fields of the standard model (SM) in numerous ways. Such a coupling, sometimes called a 'portal', is modelled by the addition of a parameterized interaction term to the SM Lagrangian[25,26]. In this paper, we consider linear interaction terms with the electron rest mass $m_e$ and the electromagnetic field tensor $F_{\mu\nu}$:

$$\mathcal{L}_{\text{int}} \supset \frac{\phi}{\Lambda_\gamma} \frac{F_{\mu\nu}F^{\mu\nu}}{4} - \frac{\phi}{\Lambda_e} m_e \bar{\psi}_e \psi_e, \quad (2)$$

where $\psi_e$, and $\bar{\psi}_e$ are the SM electron field and its Dirac conjugate, respectively, and $\Lambda_\gamma$ and $\Lambda_e$ parameterize the coupling. Specific types of scalar dark matter, such as the hypothetical moduli and dilaton fields motivated by string theory, have couplings to the quantum chromodynamics part of the SM as well[27-29].

The addition of the terms in equation (2) to the SM Lagrangian entails changes in the fine structure constant $\alpha$ and the electron rest mass $m_e$ (refs. [2,3]). The apparent variation in these fundamental constants, in turn, changes the lattice spacing and electronic modes of a solid, driving changes in its size $l$ and refractive index $n$:

$$\frac{\delta l}{l} = -\left(\frac{\delta\alpha}{\alpha} + \frac{\delta m_e}{m_e}\right), \quad (3)$$

$$\frac{\delta n}{n} = -5 \times 10^{-3}\left(2\frac{\delta\alpha}{\alpha} + \frac{\delta m_e}{m_e}\right), \quad (4)$$

where $\delta x$ denotes a change in the parameter $x$: $x \to x + \delta x$. Equations (3) and (4) hold in the adiabatic limit, which applies for a solid with a mechanical resonance frequency much higher than $\omega_\phi$ (the driving frequency)[16,28,30].

Laser interferometers for gravitational-wave detection are modified Michelson interferometers with very high sensitivity to differential changes in the optical path length of their arms. The thin cylindrical beam splitter in such an instrument interacts asymmetrically with light from the two arms, as the front surface has a 50% reflectivity and the back surface has an anti-reflective coating. Therefore, a change in the size ($\delta l$) and index of refraction ($\delta n$) of the beam splitter affects the two arms differently and produces an effective difference in the optical path lengths of the arms $L_{x,y}$

$$\delta(L_x - L_y) \approx \sqrt{2}\left[\left(n - \frac{1}{2}\right)\delta l + l\delta n\right], \quad (5)$$

This expression includes a correction to equation (17) in ref. [16]. In addition, a geometrical correction ($\approx$6.4%) from Snell's law is applied to equations (5) and (6) for calculating the results below.

The mirrors in the arms of gravitational-wave interferometers would also undergo changes in their size and index of refraction, but as the wavelength of the dark matter field is much greater than the distance between the arm mirrors ($\lambda_\phi/L \gtrsim 10^3$) for all frequencies of interest here, and because the mirrors have roughly the same thickness, the effect is almost equal in both arms and therefore does not produce a dominant signal.

The interferometer most sensitive to potential dark matter signals is the GEO600 detector, as it has the highest sensitivity to optical phase differences between the two arms. The squeezed vacuum states of light currently used in this instrument allow for a world-record quantum noise reduction of 6 dB (ref. [31]). Although other gravitational-wave detectors (LIGO/Virgo) are more sensitive to gravitational waves through the use of longer arms, including Fabry–Pérot cavities, these do not boost their sensitivity to signals

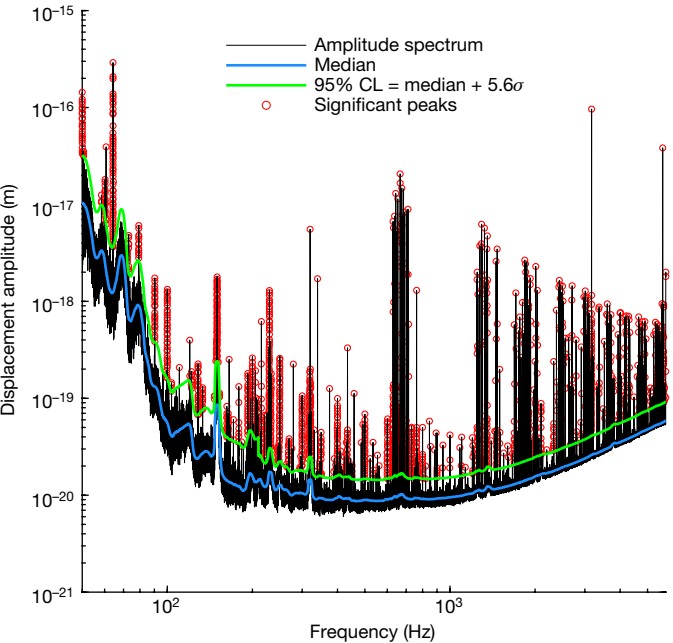

**Fig. 1 | A typical amplitude spectrum produced with frequency bins that are tuned to the expected dark matter linewidth using the modified LPSD technique.** The black line indicates the amplitude spectrum. The noise spectrum was estimated at each frequency bin from neighbouring bins to yield the local noise median (blue) and 95% confidence level (CL, green). Peaks (red) above this confidence level were considered candidates for dark matter signals and analysed further.

induced at the beam splitter, so their relative sensitivity to scalar dark matter is lower[16].

From equations (1)–(5), it follows that an oscillating scalar dark matter field is expected to produce a Doppler-shifted and Doppler-broadened signal in the GEO600 interferometer of the form

$$\delta(L_x - L_y) \approx \left(\frac{1}{\Lambda_\gamma} + \frac{1}{\Lambda_e}\right)\left(\frac{nl\hbar\sqrt{2\rho_{\text{local}}}}{m_\phi c}\right)\cos(\omega_{\text{obs}}t), \quad (6)$$

where we have neglected the contribution of the refractive index changes to the signal, as it is three orders of magnitude smaller than that of the size changes. Given this prediction, we can examine the data from the interferometer for the presence of such signals and, if none are found, place upper limits on the mass and coupling constants of scalar field dark matter.

## Results

The GEO600 interferometer[32] has been in joint observing runs with the Advanced LIGO detectors since 2015, primarily to look for gravitational waves. We performed spectral analysis on seven $T \sim 10^5$ s segments of strain data from the GEO600 interferometer (acquired in 2016 and 2019) using a modified version of the logarithmic power spectral density (LPSD) technique[33], which was designed to produce spectra with logarithmically spaced frequencies. Using this algorithm to perform discrete Fourier transforms (DFTs) with a frequency-dependent length, we created spectra in which each frequency bin was made to have a width equal to the Doppler-broadened linewidth of potential signals from scalar field dark matter in a galactic halo. A typical spectrum created using this approach is shown in Fig. 1. This method yields, in theory, the maximum attainable signal-to-noise ratio (SNR), given a certain amount of data (see Methods)[21,34].

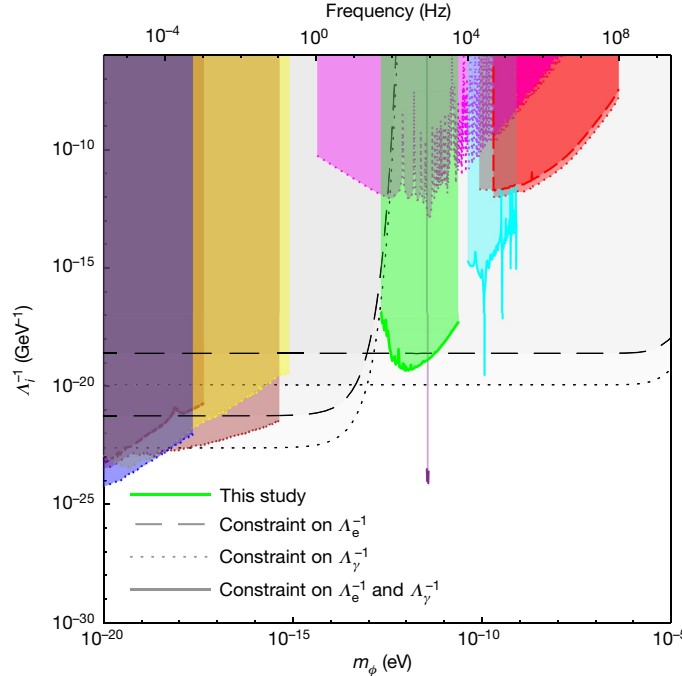

Fig. 2 | **Constraints on the coupling parameters $\Lambda_\gamma$ and $\Lambda_e$ as a function of the mass of the field $m_\phi$, for scalar field dark matter as in the basic scalar scenario.** Dashed lines represent constraints on the electron coupling $\Lambda_e$ and dotted lines represent constraints on the photon coupling $\Lambda_\gamma$, at the 95% confidence level. The green region denotes the parameter space excluded in the current study through the spectral analysis of data from the GEO600 gravitational-wave detector. Other coloured regions indicate parameter spaces excluded through previous direct experimental searches: Hees et al.[6] (blue), Van Tilburg et al.[5] (yellow), Kennedy et al.[45] (brown), Aharony et al.[38] (magenta), Branca et al.[40] (purple), Savalle et al.[8] (cyan) and Antypas et al.[39] (red). The black curves and grey regions correspond to previous constraints from 'fifth-force' searches/tests of the equivalence principle; to our knowledge, the most stringent such constraints for this dark matter scenario are from the MICROSCOPE experiment[7,43] (lower curves at low mass) and the Cu/Pb torsion pendulum experiment performed by the Eöt-Wash group[26,37,41] (at higher masses).

A matched filtering approach is not feasible, as the phase of the signal varies stochastically.

We analysed the amplitude spectra of all seven strain data segments for the presence of dark matter signals by looking for significant peaks in the underlying noise. Peaks were considered candidates when there was a less than 5% probability of the local maximum being due to noise, where we compensated for the look-elsewhere effect using a large trial factor ($\sim 10^6$).

This analysis found approximately $10^4$ peaks above the 95% confidence level ($>5.6\sigma$), where the total error includes a frequency-dependent amplitude calibration error of up to 30% inherent to GEO600 data[35]. The frequency and amplitude stability of the peaks in time was then evaluated by cross-checking all candidates between spectra. Candidate peaks were rejected if their centre frequencies differed between spectra by more than the Doppler shift expected from the motion of the Earth around the Sun through a galactic dark matter halo[36]. Peaks were also rejected if their amplitude changed significantly ($\gtrsim 5\sigma$) between spectra.

Using this procedure, we eliminated all but 14 candidate peaks, where the vast majority ($>99\%$) of peaks were rejected because they did not appear in all datasets within the centre frequency tolerance.

These 14 candidate peaks were subjected to further analysis to investigate whether their properties matched that of a dark matter signal. Thirteen of the peaks were found to have insufficient width to be caused

by dark matter ($\Delta f_{peak}/\Delta f_{DM} \lesssim 10$, see Methods). The remaining candidate peak had sufficient frequency spread to be due to dark matter, but additional analysis showed that this signal has a coherence time much greater than that expected for a galactic halo dark matter signal of that frequency ($\tau_c^{peak}/\tau_c^{GH} > 10$, see Methods). This leaves open the possibility of the signal being due to scalar dark matter gravitationally bound to Earth, such as in a relaxion halo. However, additional investigations revealed that this signal was not present in data acquired with independent electronics, whereas the noise and the other signals from the interferometer were. The peak was, therefore, rejected and is suspected to be an artefact from a timing signal in the main data acquisition electronics (see Methods for details).

Having determined that all major peaks in the amplitude spectrum are not caused by scalar field dark matter, we can set constraints on the parameters of such dark matter at a 95% confidence level (corresponding to $5.6\sigma$ above the noise floor), using equation (6). We apply our results to three different scalar dark matter scenarios considered in the literature:

Basic scalar (Fig. 2): the scalar field dark matter is assumed to interact with the SM as given by the terms in equation (2) and is further assumed to be homogeneously distributed over the Solar System with a density of $\rho_{GH} = 0.4$ GeV cm$^{-3}$, as in the standard galactic dark matter halo model[20].

Dilaton/modulus (Fig. 3, left): in addition to the coupling to the electromagnetic sector as in equation (2), the field is assumed to have couplings to the quantum chromodynamics sector and the coupling to the gluon field is assumed to be dominant[27–29,37]. The local dark matter density is taken to be $\rho_{GH}$. Compared with the basic scalar, this scenario is subject to additional limits from tests of the equivalence principle, but is equally constrained by our result and those of other direct searches.

Relaxion halo (Fig. 3, right): in this scenario, the scalar field effectively couples to the SM as in the dilaton/modulus scenario, but these couplings arise through mixing with the Higgs boson[22,23]. The local dark matter density in this scenario is taken to be dominated by a relaxion halo gravitationally bound to Earth, which leads to a local overdensity that depends on the mass of the field and reaches values of up to $\rho_{local}/\rho_{GH} = 10^{11}$ (ref. [24]) for the mass range constrained in this work.

For each scenario, we set constraints on the electron and photon coupling parameters $\Lambda_\gamma$ and $\Lambda_e$, respectively, as a function of the mass of the field $m_\phi$ (where, for each coupling constant, we assume the other to be zero); the constraints are plotted in Figs. 2 and 3, together with previous upper limits. For the relaxion halo scenario, we assumed a mass-dependent halo density, as described in ref. [24].

Constraints from other direct experimental dark matter searches include those from various atomic spectroscopy experiments[5,6,38,39], a search using an optical cavity[8] and a resonant mass detector[40]. Tests of the equivalence principle using, for example, torsion balances[37,41,42] have also been used to set constraints on the parameters of undiscovered scalar fields; these bounds assume that the scalar field manifests as a 'fifth force' and is sourced by a test mass (for example, the Earth)[7,26,43]. The constraints on scalar fields inferred from these experiments generally depend on the composition and topography of the test masses and are independent of the local dark matter density.

## Conclusions

In this paper, we presented a search for signals of scalar field dark matter in the data of a gravitational-wave detector. Scalar field dark matter would cause oscillations of the fine structure constant and electron mass, which—in turn—drive oscillations of the size and index of refraction of the beam splitter in an interferometer. This would therefore produce an oscillatory signal in a gravitational-wave detector at a frequency set by the mass of the dark matter particle.

As very extensive classical noise mitigation is used in gravitational-wave detectors, quantum technologies such as squeezed light can provide a large increase in sensitivity. Such technologies facilitate measurements

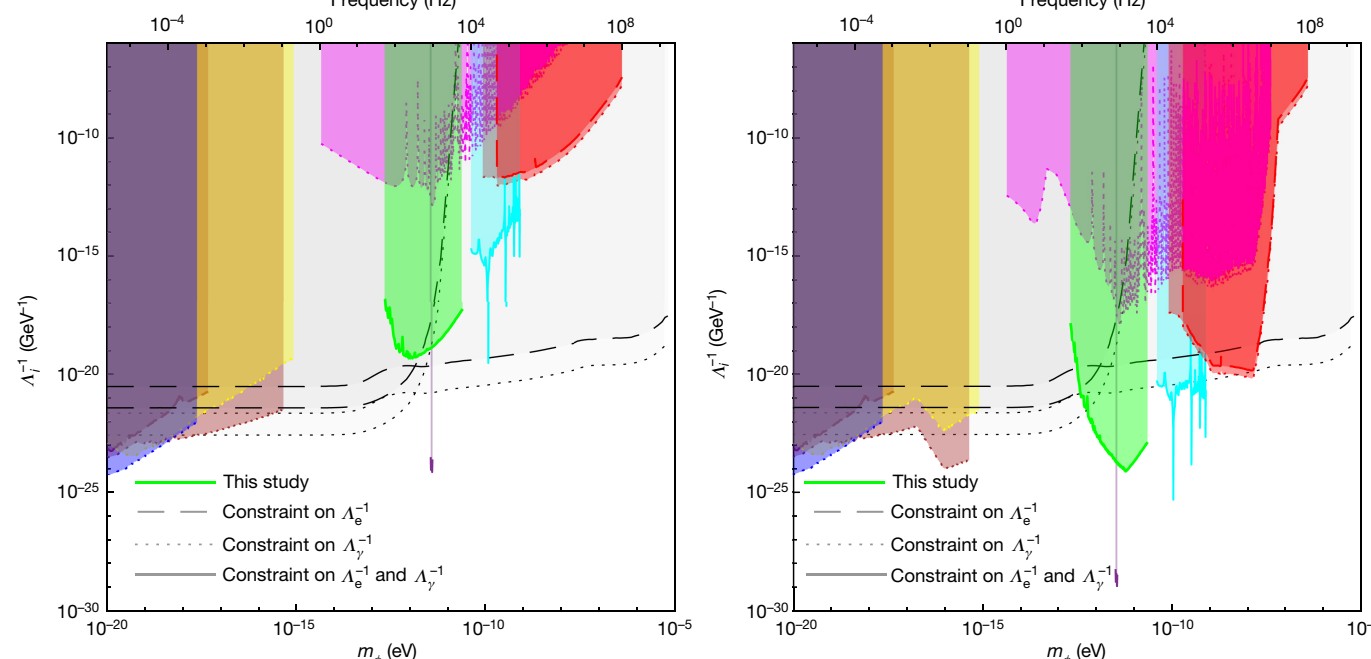

**Fig. 3 | Constraints on the coupling parameters $\Lambda_\gamma$ and $\Lambda_e$ as a function of the mass of the field $m_\phi$, for scalar field dark matter as in the dilaton/modulus scenario and the relaxion halo scenario.** The dilaton/modulus scenario is shown on the left and the relaxion halo scenario is shown on the right. Dashed lines represent constraints on the electron coupling $\Lambda_e$ and dotted lines represent constraints on the photon coupling $\Lambda_\gamma$, at the 95% confidence level. The green region denotes the parameter space excluded in the current study through the spectral analysis of data from the GEO600 gravitational-wave detector. Other coloured regions indicate parameter spaces excluded through previous direct experimental searches[5,6,38–40,45,46] (see caption of Fig. 2). The black curves and grey regions correspond to previous constraints from

'fifth-force' searches/tests of the equivalence principle; to our knowledge, the most stringent such constraints for this dark matter scenario are from the MICROSCOPE experiment[26,43] (lower curves at low mass) and the Be/Ti torsion pendulum experiment performed by the Eöt-Wash group[26,42] (at higher masses). The constraints for the relaxion halo scenario from direct experimental searches have been obtained by rescaling the originally reported constraints to account for the mass-dependent local overdensities as proposed in ref. [24]. This produces novel constraints not reported before for relaxion halo dark matter from the results of refs. [5,6,38,40,45]. The fifth force/equivalence principle constraints are independent of the local dark matter density and are, thus, unchanged.

beyond the shot-noise quantum limit and yield unprecedented sensitivity to scalar field dark matter in a wide mass range.

In addition, by tuning the frequency bin widths to the expected dark matter linewidth, our spectral analysis method improves on the analyses used in previous work that set constraints on dark photons using data from gravitational-wave detectors and other searches for scalar fields in frequency space. In contrast to these other efforts, the spectral analysis presented here yields the optimal SNR for potential dark matter signals across the full frequency range.

We excluded the presence of such signals in the data of the GEO600 gravitational-wave detector, thereby setting new lower limits on dark matter couplings at up to $\Lambda_\gamma, \Lambda_e = 3 \times 10^{19}$ GeV for dark matter masses between $10^{-13}$ and $10^{-11}$ eV. These constraints improve on the current limits in the mass range obtained with atomic spectroscopy experiments by more than six orders of magnitude and are up to four orders of magnitude more stringent than previous bounds from tests of the equivalence principle for some dark matter scenarios.

Tighter constraints on scalar field dark matter in various mass ranges can be set in the future using yet-to-be-built gravitational-wave detectors or other similar precision interferometers. Using the same methods as in this work, these instruments would allow new limits to be set across their characteristic sensitive frequency range. Moreover, by slightly modifying the optics in such interferometers—for example, by using mirrors of different thicknesses in each interferometer arm—their sensitivity to scalar field dark matter could be improved further[16]. Through the reduction of losses, quantum technologies such as squeezed light are also expected to improve, allowing for increasing noise mitigation[44]. These and other future technological advances make precision

interferometers operating beyond quantum limits indispensable tools for dark matter detection and fundamental physics in general.

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

# Methods

## Spectral estimation

Spectral analysis was performed using a modified version of the LPSD technique[33]. This technique is designed to produce spectral estimates with logarithmically spaced frequencies and therefore enables the production of spectral estimates with a frequency-dependent bin width. Using this technique, we subdivided the ~$10^5$ s data segments into

$$N_f = \left\lfloor \frac{T - \tau_{\mathrm{coh}}(f)}{\tau_{\mathrm{coh}}(f)(1-\xi)} + 1 \right\rfloor \qquad (7)$$

smaller overlapping subsegments $S_f^k(t)$ with a length equal to the expected coherence time $\tau_{\mathrm{coh}}(f)$, of a dark matter signal at a frequency $f$, where $\xi \in [0,1]$ is the fractional overlap of the subsegments and $k \in [1, N_f]$. As the expected coherence time and linewidth is frequency dependent, this subdivision is unique for every frequency of interest. After subdivision, the subsegments were multiplied with a Kaiser window function $W_f(t)$ and subjected to a DFT at a single frequency:

$$a^k(f) = \sum_{t=0}^{T_{\mathrm{DFT}}} W_f(t) S_f^k(t) e^{2\pi i f t}, \qquad (8)$$

with $T_{\mathrm{DFT}} = \tau_{\mathrm{coh}}(f)$, where $a^k(f)$ is the complex spectral estimate at frequency $f$ for the $k$th subsegment. Frequency points are chosen by dividing the interval between the chosen minimum frequency (50 Hz) and the Nyquist frequency (≈8.2 kHz) by the dark matter linewidth and then rounding the resulting number of bins to the nearest integer to set the final frequency points and bin widths. The absolute squared magnitudes $|a^k(f)|^2$ are averaged over the subsegments to obtain the power spectrum

$$P(f) = \frac{C}{N_f} \sum_{k=1}^{N_f} |a^k(f)|^2, \qquad (10)$$

where $C$ is a normalization factor. The spectra used in the analysis were made with a bin width equal to the expected linewidth of galactic dark matter ($\Delta\omega/\omega \approx 10^{-6}$, see ref. [21]). The amplitude spectrum $A(f) = \sqrt{P(f)}$ created in this way comprises approximately $5 \times 10^6$ frequency bins between 50 Hz and 6 kHz.

The SNR for galactic dark matter signals in such a spectrum is optimal given a certain amount of data (see Sec. 'Validation of methods') and can only be further improved by analysing more data. Additional data would allow for more averaging, which decreases the variance of the spectrum as the square root of the amount of data, such that the sensitivity approaches the noise floor. The noise floor can be lowered using longer DFT lengths at the cost of reduced SNR, but this is subject to severely diminishing returns; the sensitivity can only be improved by a factor proportional to the fourth root of the amount of data needed[21] (and the computation time scales with the product of DFT length and the amount of data[33]). Computation times for the spectra used in this work are approximately 10 s per frequency bin for each ~$10^5$ s dataset, or about $10^4$ CPU hours per spectrum.

## Estimation of noise statistics

The local noise parameters were estimated at every frequency bin from $w = 5 \times 10^4$ neighbouring bins. This method allows the underlying noise distribution to be estimated in a way that is independent of narrow ($\ll w$) spectral features (such as those due to mechanical excitation of the mirror suspensions), under the assumption that the underlying noise spectrum is locally flat (that is, the autocorrelation length of the noise spectrum is assumed to be $\gg w$). Therefore, the choice of $w$ represents a trade-off between erroneously assuming instrumental spectral artefacts or signals to be features of the underlying noise spectrum versus erroneously assuming features of the underlying noise spectrum to be instrumental spectral artefacts or signals.

## Follow-up analysis of candidates

As mentioned above, 14 candidate peaks remained after cross-checking spectra taken at different times. Thirteen of these peaks were found to have insufficient width to be dark matter signals. Further investigation of each of these candidates found that shifting the bin centre frequencies by an amount much smaller than the expected linewidth of dark matter signals of that frequency and amplitude and recomputing the spectra did not reproduce the peak. Additional work revealed that these 13 candidate peaks were not present in spectra created using the same data and the same LPSD algorithm implemented in a different programming language, whereas the noise floor and other spectral features were reproduced identically. Therefore, these peaks are probably artefacts of the numerical implementation of the LPSD technique.

The coherence time of the single remaining candidate peak was investigated by evaluating its height in the amplitude spectrum as a function of the DFT length (see 'Validation of methods'). The height of the peak did not decrease for DFT lengths more than an order of magnitude greater than the expected dark matter coherence time, showing a coherence time much greater than that expected for a galactic dark matter signal of that frequency. To find the origin of the signal and to check whether it could be due to the theoretically more coherent relaxion halo dark matter, we performed spectral analysis on data acquired on an independent data acquisition system. The signal was not present in these data, whereas both noise and other signals from the interferometer were. This fact, in combination with high-resolution ($\Delta f/f \sim 10^{-7}$) spectra revealing that the frequency at which the peak occurs is very close to and indistinguishable from an integer ($f_{\mathrm{peak}} = 224 \pm (2 \times 10^{-5})$ Hz), implies that the signal is most likely an artefact of a timing signal in the main data acquisition electronics.

## Validation of methods

To validate several aspects of our analysis methods, we simulated dark matter signals and injected them into sets of real and simulated data. The dark matter signals were created by superposing approximately $10^2$ sinusoids at frequencies linearly spaced around a centre frequency (the simulated Doppler-shifted dark matter Compton frequency), where the amplitude of each sinusoid is given by the quasi-Maxwellian dark matter line shape proposed in ref. [21] scaled by a simulated dark matter coupling constant; the relative phases of the sinusoids are randomized to capture the thermalization of the scalar field dark matter.

To test the spectral estimation, signal search and candidate rejection, a blind injection of simulated dark matter signals into several GEO600 datasets was performed, where the frequency, amplitude and number of signals was masked to the authors. All injected signals were recovered at their Compton frequency and at an amplitude corresponding to the hypothetical coupling constant and were subsequently identified through cross-checks between spectra as persistent candidate dark matter signals.

The previously proposed[17,21] and herein used condition of setting the frequency bin widths equal to the expected dark matter linewidth for attaining optimal SNR was tested using simulated dark matter signals as well. Mock dark matter signals and monochromatic sine signals were injected into real GEO600 data and Gaussian noise and spectra were made for which the width of the frequency bins $\Delta f_{\mathrm{bin}}$ (and, correspondingly, the length of the DFTs $T_{\mathrm{DFT}}$) was varied over four orders of magnitude. The recovered amplitude of signals injected into GEO600 data in spectra created using the LPSD algorithm is plotted in Extended Data Fig. 1 (left). This shows that the recovered amplitude of signals starts to decrease as the DFT length exceeds the coherence time (a monochromatic sine has infinite coherence time) and validates the rejection of the remaining candidate signal above, as its amplitude was found to be roughly constant for $T_{\mathrm{DFT}}/\tau_c > 10$. The recovered SNR of signals injected into Gaussian noise in spectra created using Welch's method[47] is plotted in Extended Data Fig. 1 (right), which confirms that the SNR is maximal when the frequency bin width is roughly equal to the

full width at half maximum $\Delta f_{DM}$ of the spectral line shape of the signal. This is a consequence of the aforementioned decrease in recovered amplitude for smaller bin widths and the scaling of white Gaussian noise.

## Data availability

The upper limit data in Figs. 2 and 3 and intermediate results, such as the spectrum in Fig. 1, are available from the corresponding author on request. The raw data used for the full analysis comprise about 80 GB and are available from the corresponding author on reasonable request.

## Code availability

The code used for this analysis has been released and can be found at https://github.com/philrelton/Scalar-Dark-Matter-LPSD.

47.    Welch, P. The use of fast Fourier transform for the estimation of power spectra: a method based on time averaging over short, modified periodograms. *IEEE Trans. Audio Electroacoust.* **15**, 70–73 (1967).

**Acknowledgements** We thank Y. Stadnik and Y. Michimura for discussion and comments on this work; D. Macleod and P. Hopkins for programming assistance; and M. Tröbs and G. Heinzel for permission to use their LPSD code. We are grateful for support from the Science and Technology Facilities Council (STFC), grants ST/T006331/1, ST/I006285/1 and ST/L000946/1, the Leverhulme Trust, grant RPG-2019-022, and the universities of Cardiff and Glasgow in the UK, the Bundesministerium für Bildung und Forschung, the state of Lower Saxony in Germany, the Max Planck Society, Leibniz Universität Hannover and Deutsche Forschungsgemeinschaft (DFG, German Research Foundation) under Germany's Excellence Strategy EXC-2123 QuantumFrontiers 390837967. This work was also partly supported by DFG grant SFB/Transregio 7 on Gravitational Wave Astronomy. We further thank W. Grass for his years of expert infrastructure support for GEO600. This document has been assigned LIGO document number LIGO-P2100053.

**Author contributions** S.M.V. and P.R. analysed the data and compiled the results; H.G. instigated this work and S.M.V. and H.G. wrote the manuscript; V.R. gave critical input to the analysis. C.A., J.L. and K.D. led the GEO600 instrument group during the period when data for this work were acquired. F.B., A.B, S.D., H.L., N.M., S.N., E.S., B.S., K.A.S., M.B., V.K., M.W. and H.W. worked on the instrument in different capacities required to achieve sensitivity and extended run duration; B.W. provided laser expertise and H.V. and M.M. built the squeezed-light source.

**Competing interests** The authors declare no competing interests.

**Additional information**
**Correspondence and requests for materials** should be addressed to Hartmut Grote.

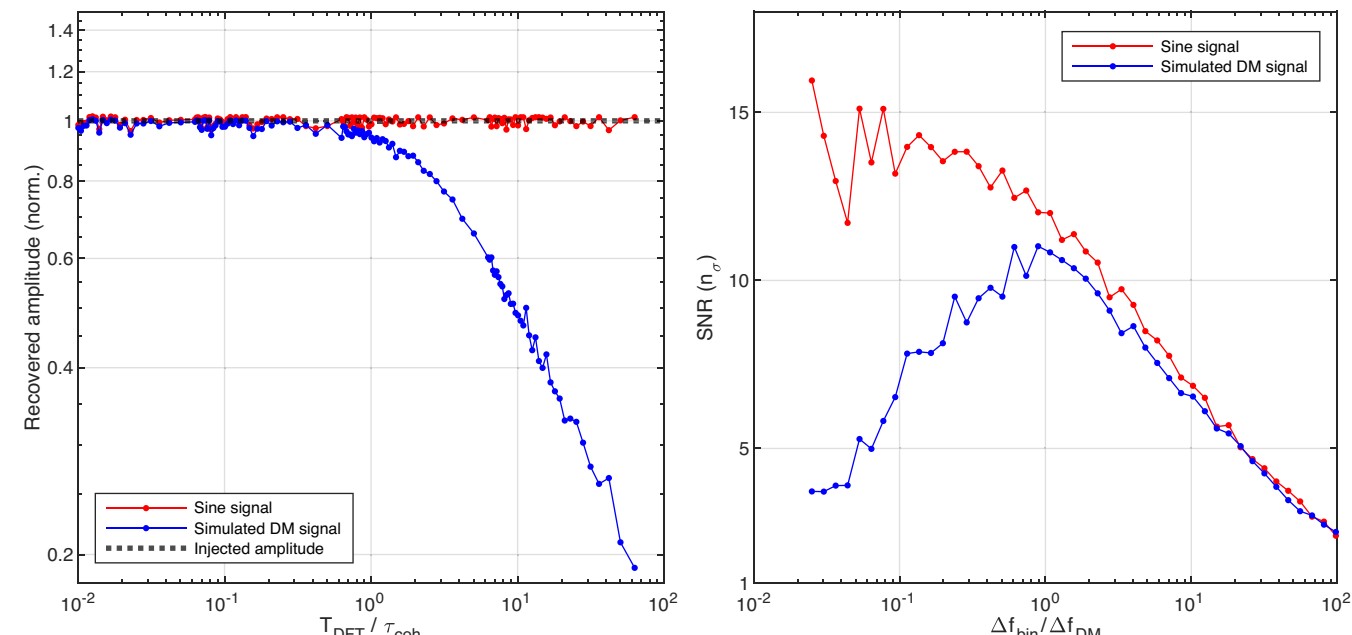

**Extended Data Fig. 1 | The spectral amplitude and SNR of a simulated dark matter signal and monochromatic sine wave as recovered from spectra created using different frequency bin widths ($\Delta f_{bin} = 1/T_{DFT}$).** The plotted recovered amplitude is normalized by the injected amplitude. The SNR ($n_\sigma$) is measured as the difference between the signal amplitude and the noise amplitude divided by the standard deviation of the noise. The appearance of a maximum for the SNR as shown on the right is a direct consequence of both the decrease in the recovered amplitude of signals with limited coherence (as shown on the left) and the scaling of white Gaussian noise with increasing integration time. The plot on the left was produced by injecting a simulated dark matter (DM) signal and a perfect sine into a segment of GEO600 data and creating spectra using the modified LPSD technique described above. The plot on the right was made by injecting the same signals into white Gaussian noise and creating spectra using Welch's method. Note that, for any single bin and for equal $T_{DFT}$, the spectral estimate obtained with the LPSD method (equation (8)) is mathematically equal to that obtained with Welch's method.