## [Peer Review File · Nature]

Manuscript Title: Direct limits for scalar field dark matter from a gravitational-wave detector

Reviewer Comments & Author Rebuttals

Reviewer Reports on the Initial Version:

Referee #1 (Remarks to the Author):

This manuscript describes the result of direct search for scalar field dark matter. This is a novel and original work showing the first demonstration of scalar field dark matter search using a sensitive laser interferometric gravitational-wave antenna. Moreover, the upper limit was improved from previous direct searches by more than six orders of magnitude. The analysis method, interpretation, and verification procedure look reasonable. The manuscript is written in a clear format with appropriate length. I suggest to accept this manuscript for publication.

I have a few minor questions:

* Frequency and amplitude consistency is tested in the veto of the candidate peaks. I am curious about the effect of scalar field dark matter on the frequency standard used in this detector (or data recorder).

* I wonder why the recovered amplitude is so consistent with the injected signal on Fig. 3 even with an estimated calibration error of 30%. How do you estimate the time change in the amplitude calibration factor? How do you treat it in the injection test?

Referee #2 (Remarks to the Author):

The manuscript studies a specific (hypothetical) form of dark matter (DM), namely the one where a weakly coupled, very low mass ($\ll 1$ eV) oscillating scalar field constitutes a majority of the DM energy density. By utilizing the data collected by the gravitational wave detector GEO 600, the authors report the results of the first direct search for scalar field DM.

In the beginning of the manuscript, the authors make a specific choice by assuming the DM is a coherently oscillating scalar field which couples linearly to the electron and the electromagnetic field tensor, while other types of couplings are neglected; see Eqs. (1) and (2). The authors then base their analysis on the assumption – derived in their Ref. [16] by one of the authors – that the aforementioned couplings change the fine structure constant and the electron mass, which in turn changes the size and refractive index of the detector's beamsplitter, leading to a potentially detectable signal.

When deriving what this signal would be, not much details are given, and the errors / subdominant factors associated to Eqs. (3)-(4) and e.g. the "geometrical correction factor" in footnote 1 are not discussed to a sufficient accuracy (i.e. quantitatively). Also, the authors do not show how different quantities such as the refractive index in Eqs. (3) and (4) depends on the assumed couplings between DM and the Standard Model fields. This makes it difficult to understand why the reported constraints in different panels of Fig. 2 are different from each other, as naively one would think from Eq. (4) that the constraints depend on the couplings in the same way. The authors also do not explain nor give a reference on how exactly the observed DM frequency w_{obs} is modulated by the motion of the Earth with respect to the galactic DM halo, which makes it difficult to understand how accurate the results are.

Based on their analysis, described in Secs. III and V, the authors report that they have not

detected any signals they could associate to DM. Based on this, the authors claim they can derive constraints on the DM-electron and DM-electromagnetic field tensor couplings as a function of the DM scalar field mass, however these constraints also depend on the local DM density which is currently unknown (and the estimates vary by orders of magnitude), as the authors correctly state in the end of Sec. III. They also say their results lead to conservative limits while assuming a larger local DM density would lead to more stringent constraints. It is not clear from the figure, however, how the other constraints (those not reported for the first time here) would change if a different local DM density was assumed. Therefore, assessing how stringent the reported constraints actually are and how they relate to previous results reported in the literature is difficult.

Without taking stance at this point on how the data analysis was conducted and whether the results seem correct or not, I can say that in my opinion the manuscript suffers from a clear shortcoming. That is, while the idea of searching for scalar field DM with gravitational wave detectors and the results presented in the manuscript are in principle interesting, in my opinion the starting point (the DM is a coherently oscillating scalar field which couples only linearly and only to the electron and the electromagnetic field tensor) is much too specific for the results to be published in Nature. I do not see why the manuscript would meet all the publishing criteria, in particular the one that requires published articles to be "of extreme importance to scientists in the specific field". In my opinion a specialist journal would be a much more suitable place for this manuscript. Therefore, I recommend rejecting the article.

Referee #3 (Remarks to the Author):

The authors searched for peaks in the GEO-600 dataset, which would be predicted by coherent scalar field dark matter according to a derivation in Ref. [16]. This derivation relates the coherent oscillations in the electromagnetic fine structure constant and the electron rest mass caused by ultralight scalar dark matter (cf. Ref. [4]) to the size and refractive length of a solid [16]. One such a solid could be the beamsplitter in a gravitational wave interferometer experiment, and following [16] this would mean that the presence of ultralight scalar dark matter its size and refractive length would change. As the GEO-600 beamsplitter has a slight asymmetry in its interactions with the interferometer arms, this gives rise to phase differences which can in principle be analyzed to look for evidence of dark matter.

A challenge of this methodology is that the phase of the signal varies stochastically. The authors describe a procedure for the identification of relevant frequency peaks (some questions below). The authors do not find any evidence for coherent scalar dark matter and proceed to set constraints.

The constraints derived in this paper are, besides barely being competitive (not competitive in the case of the photon coupling) with existing constraints from terrestrial Equivalence Principle tests, also far from surprising, as they follow from the properties of the experiment and were already anticipated in Ref. [16] (Fig. 3).

I am also concerned that the competitiveness of the constraints depends on the approximations made in deriving (3) and (4), which are not clearly presented in this work. In addition, my own estimation of the Eot-Wash constraints in this figure comes out stronger.

In this light the sentence "examining data from the GEO 600 detector for the presence of such a signal therefore allows us to set constraints on the properties of scalar dark matter." appears to be somewhat misleading.

Although the effect studied in this paper is interesting, it was proposed in earlier work. The main novel contribution made in this work is the methodology used to analyze the data, but this is in my eyes not sufficiently original nor clearly presented to warrant publication in Nature. Nevertheless, the work could be improved by a stronger focus on the original aspects of the data

analysis and a prognosis for its use in other experimental setups to set competitive limits, including for example a justification of the selection and rejection method of the peaks.

Further questions and comments:

- Could the authors clarify the probabilities in the following two sentences: "Peaks were considered candidates when there was a less than 1 percent probability of the local maximum being due to noise", and "This analysis found $\sim 10^4$ peaks above the 95 percent confidence level ($>5.6\sigma$)"?
- Selection and rejection of peaks: can the authors explain why this procedure is reliable? In particular, I would like to understand the initial selection of too narrow peaks (Section V.C) in combination with stochastic noise and the requirement on centre frequencies.
- Should Rot-Wash be Eot-Wash?

Author Rebuttals to Initial Comments:

Response to Reviewer comments for submission ‘Direct limits on scalar field dark matter from a gravitational-wave detector’.

We thank the Reviewers for their time invested in reading our manuscript. Their critical remarks have instigated considerable changes to the paper, the main one being the additional interpretation of our work for various dark matter models, which has made the scope of our work more explicit.

Referee #1 (Remarks to the Authors):

This manuscript describes the result of direct search for scalar field dark matter. This is a novel and original work showing the first demonstration of scalar field dark matter search using a sensitive laser interferometric gravitational-wave antenna. Moreover, the upper limit was improved from previous direct searches by more than six orders of magnitude. The analysis method, interpretation, and verification procedure look reasonable. The manuscript is written in a clear format with appropriate length. I suggest to accept this manuscript for publication.

I have a few minor questions:

* Frequency and amplitude consistency is tested in the veto of the candidate peaks. I am curious about the effect of scalar field dark matter on the frequency standard used in this detector (or data recorder).

Response:

Effects of the scalar field on the laser itself (i.e. modulation of the laser cavity size and refractive index) are highly suppressed by the control loop that locks the laser frequency to the common-mode arm length of the long interferometer arms. Further to this, laser frequency fluctuations are highly suppressed by the near equal length of the two interferometer arms.

Regarding an influence of the scalar dark matter on the clock of the data recorder: Each cycle of oscillation of the dark matter field that can be recorded is resolved by two or more samples from the data acquisition system. Assuming the data acquisition system is timed by a crystal oscillator at the dark matter frequency range of interest (at low frequencies it is synchronised to GPS timing signals) the sampling could have a timing jitter at the same frequency as a potential dark matter signal.

However, any strain changes of the timing crystal have to be compared to the strain changes of the interferometer beam-splitter. For the latter we are sensitive to strain changes of approx. $1e-18$. Timing errors of $1e-18$ are orders of magnitude lower than the natural jitter of the timing system. Therefore, effects of the scalar field dark matter on the frequency standard are negligible.

* I wonder why the recovered amplitude is so consistent with the injected signal on Fig. 3 even with an estimated calibration error of 30%. How do you estimate the time change in the amplitude calibration factor? How do you treat it in the injection test?

Response:

We did not include the amplitude calibration error in our injection tests. The purpose of the injection was to prove that a DM signal would produce the expected lineshape in a spectrum, and would be correctly identified as a persistent candidate signal that survives through our candidate rejection process.

Referee #2 (Remarks to the Authors):

The manuscript studies a specific (hypothetical) form of dark matter (DM), namely the one where a weakly coupled, very low mass ($\ll 1$ eV) oscillating scalar field constitutes a majority of the DM energy density. By utilizing the data collected by the gravitational wave detector GEO 600, the authors report the results of the first direct search for scalar field DM.

In the beginning of the manuscript, the authors make a specific choice by assuming the DM is a coherently oscillating scalar field which couples linearly to the electron and the electromagnetic field tensor, while other types of couplings are neglected; see Eqs. (1) and (2).

The authors then base their analysis on the assumption – derived in their Ref. [16] by one of the authors – that the aforementioned couplings change the fine structure constant and the electron mass, which in turn changes the size and refractive index of the detector's beamsplitter, leading to a potentially detectable signal.

When deriving what this signal would be, not much details are given, and the errors / subdominant factors associated to Eqs. (3)-(4) and e.g. the “geometrical correction factor” in footnote 1 are not discussed to a sufficient accuracy (i.e. quantitatively).

Response:

The relevant calculations and assumptions are given in Grote & Stadnik Phys. Rev. Research 1, 033187 – Published 19 December 2019 as cited, please see sections II and III of that article. We did not include these originally to keep the manuscript concise. We have now added two equations (now Eqs. 3 and 4) and some text to clarify the assumptions made and give some more details of the derivation. We note that the prediction that linear couplings between new scalar fields and the electron and photon

fields would produce oscillatory changes of the fine structure constant and electron mass which in turn change the size and refractive index of solids is not a novel idea by Grote & Stadnik, but is ubiquitous in literature (see e.g. Geraci et al (2019), Arvanitaki et al (2016, 2015), and various works by Stadnik & Flambaum). The geometrical correction factor due to Snell's law amounts to an expected signal reduction of 6.4%, which we have now specified as such in the manuscript instead of " $<10\%$ ".

Also, the authors do not show how different quantities such as the refractive index in Eqs. (3) and (4) depends on the assumed couplings between DM and the Standard Model fields. This makes it difficult to understand why the reported constraints in different panels of Fig. 2 are different from each other, as naively one would think from Eq. (4) that the constraints depend on the couplings in the same way.

Response:

The reported constraints on electron and photon coupling in the two panels are in fact the same, as the Referee correctly thought. We realise this was visually confusing in the original manuscript, and this has now been made more clear in the new figures and accompanying captions.

The authors also do not explain nor give a reference on how exactly the observed DM frequency w_{obs} is modulated by the motion of the Earth with respect to the galactic DM halo, which makes it difficult to understand how accurate the results are.

Response:

The modulation of the centre frequency of galactic DM, which occurs due to the motion of the earth around the sun, is on the order of $1e-7$ of the frequency (see e.g. Freese et al 2013), which is an order of magnitude smaller than our frequency resolution, and the accuracy of the DM masses for which we report the constraints is therefore limited by the latter. Note that we have now changed the wording in the manuscript as we also consider DM scenarios for which the DM centre of mass is stationary with respect to earth (in this case no modulation is expected).

Based on their analysis, described in Secs. III and V, the authors report that they have not detected any signals they could associate to DM. Based on this, the authors claim they can derive constraints on the DM-electron and DM-electromagnetic field tensor couplings as a function of the DM scalar field mass, however these constraints also depend on the local DM density which is currently unknown (and the estimates vary by orders of magnitude), as the authors correctly state in the end of Sec. III. They also say their results lead to conservative limits while assuming a larger local DM density would lead to more stringent constraints. It is not clear from the figure, however, how the other

constraints (those not reported for the first time here) would change if a different local DM density was assumed. Therefore, assessing how stringent the reported constraints actually are and how they relate to previous results reported in the literature is difficult.

Response:

We have now interpreted our results in the context of three different DM scenarios, with different assumptions for the local dark matter density. For two scenarios we assume the value of 0.4 GeV/cm^3 , which is a common estimate for the galactic halo DM, and most previous constraints were set assuming this value. For the Relaxion Halo we assume the mass-dependent density as in Banerjee et al (2020), which has been used in most other works that set constraints for this scenario (Savalle et al 2021, Antypas et al 2019). For the direct experimental searches shown in the manuscript, the constraints scale with the square root of the DM density as the field amplitude is proportional to the square root of the local DM density. We have applied this scaling to our constraints and where necessary to other constraints in our plots such that all constraints are shown for equal assumptions. The FF/EP constraints are independent of the local DM density and are therefore unchanged.

Without taking stance at this point on how the data analysis was conducted and whether the results seem correct or not, I can say that in my opinion the manuscript suffers from a clear shortcoming. That is, while the idea of searching for scalar field DM with gravitational wave detectors and the results presented in the manuscript are in principle interesting, in my opinion the starting point (the DM is a coherently oscillating scalar field which couples only linearly and only to the electron and the electromagnetic field tensor) is much too specific for the results to be published in Nature. I do not see why the manuscript would meet all the publishing criteria, in particular the one that requires published articles to be “of extreme importance to scientists in the specific field”. In my opinion a specialist journal would be a much more suitable place for this manuscript. Therefore, I recommend rejecting the article.

Response:

We do not think the model we considered in our original manuscript is specific, unless one judges the entire category of light scalar field dark matter models to be specific. We consider light scalar field DM that couples linearly to the electron and the electromagnetic field tensor, and no other couplings. We made this choice because we consider it to be the most general form of scalar field dark matter with minimal assumptions; including other couplings relies on additional assumptions which we would argue actually make the DM model more specific. This does not mean that our constraints do not apply to other DM models with different assumptions; we are

confident our results imply competitive constraints on most forms of light scalar field dark matter that couple to the electromagnetic and electron fields. We realise we failed to point this out in the original manuscript.

For this reason, we have now interpreted our experimental results in the context of dilaton, moduli and relaxation halo dark matter in our revised manuscript. We believe this has improved the article and makes the results more accessible and their impact more readily apparent.

It is not for us, (and it would seem difficult in general), to judge whether our work is “of extreme importance for scientists in the specific field”. However, we believe the value of our manuscript lies not only in reporting novel constraints on a promising DM candidate (deemed such by a significant part of the community, evidenced by the large number of recent publications on this topic), but in demonstrating the successful application of a novel experimental technique to search for scalar DM and oscillatory changes of fundamental constants in general. Broadband experimental probes for this phenomenology have for the past five years or so been limited to clock-clock and clock-cavity comparisons. The addition of (quantum-enhanced) interferometry to this field, as established in our work, allows the exploration of a wide mass/frequency range that was previously inaccessible. We would also point out that the experiment employs *the* state of the art in quantum squeezing, and this is the first fundamental physics result produced using such a high level (6 dB) of squeezing (cf. e.g. Backes, K.M., Palken, D.A., Kenany, S.A. et al., Nature 590, 238–242 (2021)).

Referee #3 (Remarks to the Author):

The authors searched for peaks in the GEO-600 dataset, which would be predicted by coherent scalar field dark matter according to a derivation in Ref. [16]. This derivation relates the coherent oscillations in the electromagnetic fine structure constant and the electron rest mass caused by ultralight scalar dark matter (cf. Ref. [4]) to the size and refractive length of a solid [16]. One such a solid could be the beamsplitter in a gravitational wave interferometer experiment, and following [16] this would mean that the presence of ultralight scalar dark matter its size and refractive length would change. As the GEO-600 beamsplitter has a slight asymmetry in its interactions with the interferometer arms, this gives rise to phase differences which can in principle be analyzed to look for evidence of dark matter.

A challenge of this methodology is that the phase of the signal varies stochastically. The authors describe a procedure for the identification of relevant frequency peaks (some questions below). The authors do not find any evidence for coherent scalar dark matter and proceed to set constraints.

The constraints derived in this paper are, besides barely being competitive (not competitive in the case of the photon coupling) with existing constraints from terrestrial Equivalence Principle tests, also far from surprising, as they follow from the properties of the experiment and were already anticipated in Ref. [16] (Fig. 3).

Response:

We find the objection that our results are “[...] far from surprising, as they follow from the properties of the experiment and were already anticipated [...]” is moot. We are sure Ref.#3 would agree that theoretical predictions cannot, in themselves, be taken as scientific proof, and that one of the purposes of experimentation and data analysis is to test the predictions given by a theory. (Another is serendipitously finding anomalies with respect to current scientific frameworks.) As such, it is of high importance that experimental results are published to make claims as to detection or elimination, and we would argue that whether these results are surprising is rather subjective and irrelevant. Concerning the comparison to constraints from FF/EP experiments, we would like to point out there is much to be said about why EP or fifth force tests are less competitive than direct searches; the constraints from these tests rely on a large number of assumptions to allow them to be applied to dark matter models.

As for ‘the challenge of this methodology’ due to the stochastic phase variations: This challenge applies to all dark matter searches assuming a virialized field. In our work we address this challenge by using a method which maximises the signal-to-noise ratio by tuning the width of frequency bins to the DM linewidth. This allows us to set more optimal constraints given the data compared to previous searches, as explained and demonstrated in Sec. V.

I am also concerned that the competitiveness of the constraints depends on the approximations made in deriving (3) and (4), which are not clearly presented in this work. In addition, my own estimation of the Eot-Wash constraints in this figure comes out stronger.

Response:

As to the concern around the approximations used, we would like to point out that all approximations are detailed in Grote & Stadnik (2019), and we did not include these to keep the manuscript concise. We summarise the relevant approximations here:

We work in the adiabatic limit for the DM-driven BS size changes. This is the limit for which the fundamental resonance frequency of the beamsplitter is much greater than the scalar DM driving frequency. Given the physical dimensions of the beamsplitter, its material properties, and its high Q-factor, any deviation from the adiabatic limit would produce an error in our result many orders of magnitude smaller than the error from other sources (such as the detector’s output calibration). However, importantly, a

deviation from the adiabatic limit would in this case enhance the potential signal, which would make our upper limits *more* competitive, not less. The only other relevant approximation we make is to neglect the contribution to the signal from changes of the refractive index, which are roughly 3 orders of magnitude smaller than the contribution from the size changes. Again, to forego this approximation would make our upper limits more competitive (but again, insignificantly so).

As to the concern with respect to the Eöt-Wash constraints, we would like to point out that the Eöt-Wash group has performed numerous torsion balance experiments, from which different constraints on dark matter have been inferred. In general, these constraints depend on the composition and topography of the masses used, and depend on the specific assumptions made for the dark matter couplings (see Wagner et al 2012, Hees et al 2018). Specifically, in those works and others, it is pointed out that the constraints from the Be/Ti torsion pendulum experiment (Schlamminger et al (2008)), depend strongly on the assumed composition of Earth and the local topography of the lab, and apply only for a scalar field which has a dominant coupling to the gluon field.

We have now updated the manuscript to include an interpretation of our results in the context of three different DM scenarios, and as can be seen in the manuscript they are subject to different constraints from fifth-force/equivalence principle experiments, which we outperform for two of our three scenarios.

In this light the sentence "examining data from the GEO 600 detector for the presence of such a signal therefore allows us to set constraints on the properties of scalar dark matter." appears to be somewhat misleading.

Response:

We have changed the wording of this sentence, to show more clearly that it is the exclusion of potential DM signals that allows us to set upper limits, it now reads: "Given this prediction, we can examine the data from the interferometer for the presence of such a signal, and if none are found place upper limits on the mass and coupling constants of scalar field DM."

Although the effect studied in this paper is interesting, it was proposed in earlier work. The main novel contribution made in this work is the methodology used to analyze the data, but this is in my eyes not sufficiently original nor clearly presented to warrant publication in Nature. Nevertheless, the work could be improved by a stronger focus on the original aspects of the data analysis and a prognosis for its use in other experimental setups to set competitive limits, including for example a justification of the selection and rejection method of the peaks.

Response:

We would like to point out that the main novel contributions of this work are to present a new physical result (a new upper limit on scalar dark matter), to demonstrate a new experimental technique that allows unprecedented sensitivity to broadband oscillatory changes of fundamental constants, and to provide an actual analysis of real data without which the former results would be meaningless. We believe the focus should be on these aspects, and that the novel analysis methodology, while important, currently receives an appropriate amount of focus in the paper. We would like to return here to our earlier point that the mere proposition of experiments and theories does not fully constitute the scientific process, and leaves it in want of experimental results and analysis thereof. One could argue that all parts of the scientific process should receive equal coverage in Nature.

The selection and rejection of peaks has been validated by performing the analysis on data sets that contain simulated signals, as detailed in Sec. V. The selection of peaks is based on fundamental statistical methods for the analysis of noisy data. In short, the theoretical prediction for the DM signal provides a prediction against which the data is compared, and the noise provides a different prediction. If the data in a certain bin, corresponding to a measurement in frequency space, is statistically distinguishable from the noise prediction, that bin is initially selected as a potential signal. Thereafter, the data in that bin is analysed further in several aspects (persistence, frequency stability, etc.) to see if the data could be explained by the presence of a DM signal as described by the theory; if not, the bin is rejected. See also our response to the query below.

Further questions and comments:

- Could the authors clarify the probabilities in the following two sentences: "Peaks were considered candidates when there was a less than 1 percent probability of the local maximum being due to noise", and "This analysis found $\sim 10^4$ peaks above the 95 percent confidence level ($\square > 5.6\sigma$)"?

Response:

Concerning the sentence: "Peaks were considered candidates when there was a less than 1 percent probability of the local maximum being due to noise";

The two hypotheses for local maxima in the data are:

Null - The peak occurs due to random fluctuations in the detector. (The frequency bin contains noise only)

Alternative - The peak occurs in part due to noise, but is dominated by the presence of an underlying source in the data. (The frequency bin contains noise + some signal)

The probability of the null hypothesis being true was determined for each frequency bin, under the assumption that the noise for each bin was described by the distribution as estimated from neighbouring bins, as explained in Secs. III and V. Concretely, a noise mean and standard deviation was estimated for each frequency bin, and then the amplitude of each bin was compared to the standard deviation and assigned a probability under the assumption that the distribution is Gaussian. Frequency bins that fell within the 95% CL (compensated for the trial factor) were rejected immediately.

Concerning the sentence: "This analysis found $\sim 10^4$ peaks above the 95 percent confidence level ($\square > 5.6\sigma$)";

As mentioned above, the 95 percent confidence level is determined to lie at a certain amplitude based on the estimated noise, under the assumption that the noise is Gaussian. We compensate for the look-elsewhere-effect by using a trial factor equal to the number of bins such that this 95 percent confidence level applies for the whole dataset collectively, not a single bin. For a Gaussian distribution, the 95% confidence level given $\sim 4.8e6$ bins(/measurements) corresponds to 5.6 standard deviations.

- Selection and rejection of peaks: can the authors explain why this procedure is reliable? In particular, I would like to understand the initial selection of too narrow peaks (Section V.C) in combination with stochastic noise and the requirement on centre frequencies.

Response:

We are not entirely sure what the Referee is asking here. Initial peaks were selected purely by their height relative to the local noise floor. This is the minimal condition of evidence of a signal in the data. If any DM signal is detectable in this parameter space then it should fall in this set of peaks. After cross-checks for persistence and frequency stability we eliminated all but 14 peaks. All but one of these peaks comprised just a single bin. The single bin peaks were further investigated at higher resolution. At this higher resolution and given the prominence of the peaks, the spectral lineshape of a real DM signal should be resolved and the peak should comprise multiple bins, but this was not the case, ruling out the possibility of these peaks being due to real DM signals.

- Should Rot-Wash be Eot-Wash?

Response:

The Rot-Wash experiment is a specific torsion balance experiment performed by the Eöt-Wash group (Smith et al 1999).

Reviewer Reports on the First Revision:

Referee #1 (Remarks to the Author):

The authors' responses to my points are satisfactory. I suggest to accept the paper for publication.

Referee #2 (Remarks to the Author):

After reading the new version of the manuscript and the authors' exhaustive reply to the reviewers' comments, I have become convinced the authors have changed the manuscript and replied to all concerns raised by reviewers to my satisfaction. In particular, I have changed my mind about the article's importance and expected impact to the scientific community – I do agree with the authors that “the value of [the] manuscript lies [...] in demonstrating the successful application of a novel experimental technique to search for scalar DM and oscillatory changes of fundamental constants in general”.

I therefore recommend the article for publication, although I have a few minor points I would like to raise here and that I would like to see corrected before publication:

- In various points, the authors say “The new constraints improve upon bounds from previous direct searches by more than six orders of magnitude and are more stringent than limits obtained in tests of the equivalence principle by up to four orders of magnitude”, however, I find this statement somewhat misleading as the new constraints are not always more stringent than those obtained from tests of the equivalence principle. Therefore, I recommend adding “[...] and are *in some cases* more stringent than limits obtained in tests of the equivalence principle by up to four orders of magnitude”.
- The different symbols (δ_l , δ_α ...) appearing in Eqs. (3)-(4) should be specified below them.
- It is not entirely clear from Figs. 2-4 which constraints are for Λ_e and which for Λ_γ . I suggest the authors choose different line elements or color schemes than those currently used so that the constraints become visibly distinct enough.
- In Conclusions, the authors say they have derived upper limits for Λ_i (sic) while I think here they mean *lower* limits.

Referee #3 (Remarks to the Author):

I have read through the author's responses but remain unconvinced about the importance of the results. Here a brief reply to the main points.

While I do (of course) agree that experimental results are important and that theoretical predictions are not by themselves sufficient, this does not contradict any point I was making, and interpreting my response as if it does is a straw man argument. What I said is that the current results are not groundbreaking because they are barely competitive with existing constraints found using different experiments, a fact that should itself not have been surprising given the previously published projections. It was known a priori that the experimental sensitivity was barely (if at all) probing novel parameter space. There was no reason to expect the experimental sensitivity would outperform the published projections, which are typically idealized scenarios.

In short, while I do not think anyone would disagree that experiments should be carried out to test theoretical predictions, in this case it was not those theoretical predictions setting the existing constraint, but another experiment. The expectation that this technique could find ultralight dark matter was therefore never on very firm footing. This does not make the analysis wrong, but it does make the result less important.

The authors claim the new experimental technique is the most important contribution of the work. But this technique was proposed earlier in a published work (Grote and Stadnik), and is thus not original; the data analysis framework does seem original. More generally, I think the paper should stand on its own and not mentioning approximations (on which the final result relies) in the paper itself makes it incomplete.

The added analysis of different dark matter models is a good addition, as it illustrates a further assumption (which does not affect other experiments): that of the local dark matter density. Alas, a local dark matter overdensity of 11 orders of magnitude is not a common assumption in dark matter models.

In conclusion, I do not believe the results to be of such importance to the field of ultralight dark matter phenomenology that it warrants publication in Nature.

Author Rebuttals to First Revision:

In response to the remaining issues raised by Referee #2 we have made the following changes:

1. Added “in some cases” and “for some DM scenarios” to sentences that describes our improvement over previous constraints, as recommended by Ref. 2, in Abstract (ln. 19), and in Conclusion (ln. 287).
2. Added “where Δx denotes a change of the parameter x : $x \rightarrow x + \Delta x$ ” below Eqs. 3-4, as suggested.
3. Changed the line renderings for the Λ_e and Λ_γ constraints in Figs. 2-4 to make it easier to differentiate between these constraints.
4. Changed “upper” to “lower” on ln 279 as suggested.